# Authentication of Coffee Blends by 16-O-Methylcafestol Quantification Using NMR Spectroscopy

**Ya-Tze Lin [1], You-Lun We [1], Ya-Min Kao [1], Su-Hsiang Tseng [1], Der-Yuan Wang [1] and Shin-Yu Chen [2,*]**

[1] Division of Research and Analysis, Taiwan Food and Drug Administration, Taipei 115209, Taiwan; dywang@fda.gov.tw (D.-Y.W.)

[2] Department of Food Science, National Pingtung University of Science and Technology, Pingtung 912301, Taiwan

* Correspondence: sychen@mail.npust.edu.tw

**Abstract:** In 2019, a coffee chain in Taiwan was found to be mixing relatively cheap Robusta beans into products marketed as 100% Arabica. Many studies show 16-OMC is a remarkable marker to distinguish Robusta from Arabica beans, and nuclear magnetic resonance (NMR) is a convenient and efficient technique for 16-OMC quantification. Here, a 500 MHz NMR was employed to determine the content of 16-OMC in coffee for adulterate evaluation. A total of 118 samples were analyzed including products from the coffee chain, raw materials (single coffee beans), and other commercial products. The contents of 16-OMC in single Robusta beans were between 1005.55 and 3208.32 mg/kg and were absent from single Arabica beans. The surveillance results indicate that 17 out of 47 blend products claiming to contain 100% Arabica had 16-OMC quantifications in the range of 155.74–784.60 mg/kg. Furthermore, all 17 products were produced by the same coffee chain. We confirmed that coffee chain adulterated Arabica with Robusta in parts of their products, which claimed to include 100% Arabica. Moreover, this work highlights the free form of 16-OMC was esterified by coffee instantly. The decomposition products of 16-OMC were observed obviously in green Robusta while the mechanisms remain unclear. Future research should focus more on these aspects to further increase our understanding of these mechanisms.

**Keywords:** adulteration; Arabica; coffee; NMR; Robusta

## 1. Introduction

Coffee is one of the most popular beverages worldwide and is a widely traded global commodity [1,2]. The most commercialized coffee species are *Coffea arabica* L. (Arabica coffee) and *Coffea canephora* Pierre ex A. Froehner (Robusta coffee) [3,4]. Arabica beans have a higher price, value, and productivity than Robusta beans because of their preferable flavor and aroma and the more difficult methods required for their growth [1,2,5,6]. However, the desire for economic gains leads to the possibility of commercial coffee beans labeled as 100% Arabica to be adulterated with Robusta [2,7].

The lipid fraction is the major research focus to distinguish Arabica and Robusta coffees [6,8–10]. Diterpenes in the lipophilic extraction of coffee are well-known compounds and markers for their authenticity, including 16-O-methylcafestol (16-OMC), kahweol, and cafestol [5,8]. The structures of these compounds are presented in Figure 1. 16-OMC is the most suitable marker for the discrimination of Arabica from Robusta. 16-OMC is present exclusively in Robusta beans but is practically nonexistent in Arabica beans, and it is fairly stable after the roasting process [2,7,8]. Kahweol is higher in Arabica but present at low levels in Robusta, and cafestol exists in both species [2,5,7]. However, 16-OMC is found in genuine Arabica coffees [1,11,12]. The reported range of 16-OMC content in green Arabica coffee by using UHPLC-MS/MS, and the range of 16-O-methylated diterpenes (16-OMD) content in green Arabica coffee oil samples is 10–260 mg/kg by using NMR [11,12]. The 16-OMD is the esterification of 16-OMC and 16-O-methylkahweol (16-OMK). Although,

16-OMC or its esterified form are present in some Arabica coffees, the level is still lower than Robusta. Economic adulteration usually adds a relatively high amount of Robusta into 100% Arabica. Therefore, 16-OMC can still be used as a marker to distinguish Arabica that is adulterated with Robusta.

R = H : Free 16-O-Methylcafestol
R = CO(CH₂)₁₄CH₃ 16-O-Methylcafestol palmitate

R = H : Free Kahweol
R = CO(CH₂)₁₄CH₃ Kahweol palmitate

R = H : Free Cafestol
R = CO(CH₂)₁₄CH₃ Cafestol palmitate

**Figure 1.** Chemical structures of 16-OMC, kahweol, and cafestol.

The German standard DIN 10779 is the first official method for authenticity testing of coffee products and is used to quantify the amount of 16-OMC by high performance liquid chromatography (HPLC) [13]. However, the HPLC method requires time-consuming sample preparation [14]. Recently, several studies have used high-resolution or low-field [1]H NMR spectroscopy to quantify 16-OMC instead of HPLC [1,2,5,7,15,16]. 16-OMC was quantified by integrating the methyl signal (H21) at 3.16 or 3.17 ppm based on the residual signal of $CDCl_3$ set to 7.26 or 7.27 ppm. The major advantages of the NMR method are its simplicity, fast sample preparation and analysis, and reproducible results [5,14]. Other than NMR and HPLC, a few techniques were applied to differentiate between Arabica and Robust coffee varieties, including gas chromatography [17], liquid chromatography-mass spectrometry [1], Raman spectroscopy [18,19], and Fourier transform infrared spectroscopy [20].

In October 2019, a coffee chain in Taiwan was shown to lack authenticity in their product, which claimed to include 100% Arabica coffee. The scandal was exposed to the media. In order to confirm the authenticity of this doubtful coffee, the Taiwan Food and Drug Administration (TFDA) immediately developed and validated the official analytical method by quantifying 16-OMC using high-field [1]H NMR spectroscopy. The TFDA consequently expanded the authenticity surveillance for the coffee that claimed 100% Arabica coffee in the market. A total of 118 samples were investigated in this study. Furthermore, free form and esterified 16-OMC as well as the decomposition products of 16-OMC were found and discussed.

## 2. Materials and Methods

### 2.1. Reagents

Chloroform-D ($CDCl_3$, 99.8%D) and pyrene ($C_{16}H_{10}$, 98%) were purchased from Sigma-Aldrich (MO, US). Standard 5 mm NMR tubes were purchased from Hilgenberg (Malsfeld, Germany). The 16-O-methylcafestol standard was purchased from PhytoLab (Vestenbergsgreuth, Germany).

### 2.2. Coffee samples

A total of 118 coffee samples were collected and assigned into five groups according to their product types. The number and type of samples included 47 single green coffee beans, five single roasted coffee beans, 52 blend roasted coffee beans, 10 ground coffees, and 4 instant coffees. Information about the samples is presented in Table 1 and Supplementary Tables S1–S4. Sample SB38~52 in Table 1 and SG1~SG46 in Supplementary Table S1 were obtained from inspections. Sample SG 47 was collected from a local coffee factory, and all others were purchased or obtained from local retailers and the market. The species of green coffee beans were confirmed using DNA identification through real-time polymerase chain reaction. The identified variety is shown in Supplementary Table S1, aside from three decaf-treated green coffee beans.

**Table 1.** Information and 16-OMC content of blend roast coffee samples.

| Sample Number | Country of Production | Labeled Species | 16-OMC (mg/kg) | SD |
|---|---|---|---|---|
| SB1 | Taiwan | 100% Arabica | nd [a] | - |
| SB2 | Taiwan | 100% Arabica | nd | - |
| SB3 | Taiwan | 100% Arabica | 426.50 | 27.50 |
| SB4 | Taiwan | 100% Arabica | 234.78 | 11.18 |
| SB5 | Taiwan | 100% Arabica | 368.75 | 12.50 |
| SB6 | Taiwan | 100% Arabica | nd | - |
| SB7 | Taiwan | 100% Arabica | nd | - |
| SB8 | Taiwan | 100% Arabica | nd | - |
| SB9 | Taiwan | 100% Arabica | nd | - |
| SB10 | Taiwan | 100% coffee | 617.61 | 66.87 |
| SB11 | Taiwan | 100% Arabica | 401.74 | 35.20 |
| SB12 | Taiwan | 100% Arabica | 155.74 | 9.43 |
| SB13 | Taiwan | 100% Arabica | 296.62 | 16.37 |
| SB14 | Taiwan | 100% Arabica | 222.62 | 31.32 |
| SB15 | Taiwan | 100% Arabica | 316.57 | 17.74 |
| SB16 | Taiwan | 100% Arabica | 207.96 | 0.83 |
| SB17 | Taiwan | 100% Arabica | 404.03 | 37.17 |
| SB18 | Taiwan | 100% coffee | 554.33 | 45.48 |
| SB19 | Taiwan | 100% coffee | 266.86 | 8.29 |
| SB20 | Taiwan | 100% Arabica | 276.30 | 12.58 |
| SB21 | Taiwan | 100% Arabica | 385.00 | 15.13 |
| SB22 | Taiwan | 100% Arabica | 247.35 | 0.74 |
| SB23 | Taiwan | 100% Arabica | 282.66 | 15.79 |
| SB24 | Taiwan | 100% Arabica | 784.60 | 60.05 |
| SB25 | Taiwan | 100% Arabica | 623.74 | 39.58 |
| SB26 | Taiwan | 100% Arabica | 722.94 | 60.31 |
| SB27 | Sweden | 100% Arabica | nd | - |
| SB28 | Italia | 100% Arabica | nd | - |
| SB29 | Italia | 100% Arabica | nd | - |
| SB30 | Colombia | 100% Arabica | nd | - |
| SB31 | Italia | Arabica | 214.56 | 10.58 |
| SB32 | Australia | 100% Arabica | nd | - |
| SB33 | Germany | 100% Arabica | nd | - |

**Table 1.** *Cont.*

| Sample Number | Country of Production | Labeled Species | 16-OMC (mg/kg) | SD |
|---|---|---|---|---|
| SB34 | Taiwan | 100% Arabica | nd | - |
| SB35 | Austria | Arabica | nd | - |
| SB36 | Australia | 100% Arabica | nd | - |
| SB37 | Taiwan | 100% Arabica | nd | - |
| SB38 | Taiwan | 100% Arabica | nd | - |
| SB39 | Taiwan | 100% Arabica | nd | - |
| SB40 | Taiwan | 100% Arabica | nd | - |
| SB41 | Taiwan | 100% Arabica | nd | - |
| SB42 | Taiwan | 100% Arabica | nd | - |
| SB43 | Taiwan | 100% Arabica | nd | - |
| SB44 | Taiwan | 100% Arabica | nd | - |
| SB45 | Taiwan | 100% Arabica | nd | - |
| SB46 | Taiwan | 100% Arabica | nd | - |
| SB47 | Taiwan | 100% Arabica | nd | - |
| SB48 | Taiwan | 100% Arabica | nd | - |
| SB49 | Taiwan | 100% Arabica | nd | - |
| SB50 | Taiwan | 100% Arabica | nd | - |
| SB51 | Taiwan | 100% Arabica | nd | - |
| SB52 | Taiwan | 100% Arabica | nd | - |

[a] nd—not detectable.

### 2.3. Sample Treatment

Approximately 30 g of coffee beans were treated with liquid nitrogen and ground to a powder using a homogenizer, which is about 4~8% of the products according to the different sizes of each package. The ground coffee beans and other powder-type products were directly extracted using a published method [7] with slight modifications. Each 0.45 g of powder (accurately weighed) was extracted with 1.5 mL of $CDCl_3$ for 15 min at 1000 rpm using a high-speed dispersing device and centrifuged at $5000\times g$ for 5 min. The supernatant was then quickly filtered through a cotton wool filter into a glass vial. Then, 750 μL of supernatant (volume accurately recorded) was transferred into a glass vial with 5 mg internal standard (pyrene) (accurately weighted). After vortexing, the sample was transferred into a 5 mm NMR tube. The extractions were performed three or more times, independently. The $^1$H NMR spectra were acquired immediately after sample treatment.

### 2.4. NMR Spectroscopy

The 500 MHz $^1$H NMR spectra were collected using a JEOL ECZ500R/S1 spectrometer running the Delta V5.3.1 software which was equipped with a 5 mm FG/RO DIGITAL AUTO TUNE probe S (NM-03822R05SS), the equipment was purchased from JEOL (Tokyo, JAPAN). The probe temperature was regulated at 30 °C. For each spectrum, 8 scans were collected using a 45° pulse angle and an acquisition time of 4 s. Free induction decays (FIDs) were zero-filled to obtain the spectra of 16,384 points. The resolution was 0.24 Hz. The chemical shift in all the spectra was reported as δ and referenced to the residual signal of $CDCl_3$ set to 7.27 ppm.

### 2.5. Quantification

The absolute concentration of 16-OMC was determined by comparing the integral area value of the methyl protons at position 21 of 16-OMC at 3.17 ppm (sum of all 16-OMC signals in this area) and that of the protons of pyrene (I.S.) at 8.09 ppm [7]. The molarity of

16-OMC was acquired using formula (1) in the software, and the concentration of the free 16-OMC was calculated using formula (2).

$$
\begin{aligned}
&\text{Molarity of 16-OMC in sample (mmol/L)} \\
&= (A_{\text{16-OMC}}/N_{\text{16-OMC}}) \times (N_{\text{I.S.}} \times W_{\text{I.S.}}/A_{\text{I.S.}} \times MW_{\text{I.S.}} \times V_S)
\end{aligned}
\tag{1}
$$

where $A_{\text{16-OMC}}$ and $A_{\text{I.S.}}$ are the areas of 16-OMC and pyrene, respectively; $N_{\text{16-OMC}}$ and $N_{\text{I.S.}}$ are the proton numbers of 16-OMC (3) and pyrene (4), respectively; $W_{\text{I.S.}}$ is the accurate weight of pyrene; $MW_{\text{I.S.}}$ is the molecular weight of pyrene (202.25 Da); and $V_S$ is the volume of the supernatant that is transferred into the internal standard (0.00075 L or the exact volume).

$$
\begin{aligned}
&\text{Concentration of the 16-OMC in sample (mg/kg)} \\
&= [(C_{\text{16-OMC}} \times V \times MW_{\text{16-OMC}}) / (1000 \times W_S)] \times 1000
\end{aligned}
\tag{2}
$$

where $C_{\text{16-OMC}}$ is the molarity of 16-OMC in the sample from the software (based on formula 1), V is the $CDCl_3$ volume used in the extraction (1.5 mL), $MW_{\text{16-OMC}}$ is the molecular weight of 16-OMC (330.46 Da), and $W_S$ is the accurate weight of the sample (g).

### 2.6. Validation

A single roast Arabica coffee bean was utilized as a blank matrix for validation studies due to the complete absence of 16-OMC. For the preparation of the spiked matrix samples, each ground Arabica bean was accurately weighed to 0.45 g before adding 22.5, 45, and 225 µL of 1000 µg/mL 16-OMC standard solution. Subsequently, the test samples were spiked in the specified 16-OMC concentration range of 50, 100, and 500 mg/kg. Each concentration was measured in five repetitions. The subsequent procedure was performed as previously described for 16-OMC quantification (Sections 2.4 and 2.5). The coefficient of variation (CV%) was assessed for the determination of reproducibility, recovery, and limits of measurements.

### 2.7. Statistical Analysis

The results are expressed as the mean ± standard deviation (SD). The mean and SD were calculated from 3 replicate values. The CV was calculated as follows: $CV = SD/Mean \times 100$ for each single case.

## 3. Results and Discussion

### 3.1. $^1H$ NMR Spectra of Coffee Extracts

The NMR spectra (chemical shift 3.0–6.4 ppm) of Robusta, Arabica, and adulterated coffee are shown in Figure 2. The proton 21 of 16-OMC as a singlet at 3.17 ppm in the Robusta is shown in (a). Otherwise, the protons 2, 1, and 18 of kahweol signals at 5.9, 6.25, and 6.3 ppm, respectively, are shown in the Arabica (c). These results correspond with those previously mentioned; 16-OMC is present exclusively in Robusta but not in Arabica, and kahweol is present in greater amounts in Arabica but is lacking in Robusta [2,5,7,8]. Moreover, an adulterated sample (Arabica adulterated with Robusta) showed signals of both 16-OMC and kahweol (b).

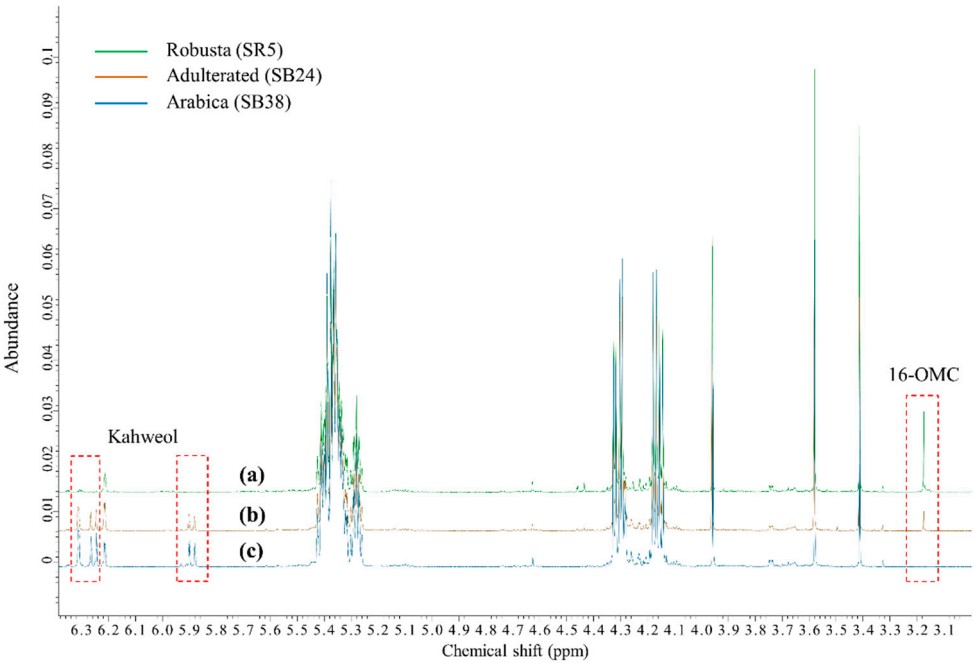

**Figure 2.** $^1$H NMR spectra of the samples that were extracted from 100% Robusta (**a**), an adulterated blend (**b**) and 100% Arabica (**c**). The expanded region of marker peaks between 3.0 ppm and 6.4 ppm.

*3.2. The Free Form and Esterified of 16-OMC*

An obvious signal of the 16-OMC free form is shown at 3.18 ppm, especially in both green and roast Robusta (Figure 3). We further analyzed the 16-OMC standard, coffee samples, and spiked the 16-OMC standard with coffee samples (Figure 4). The signal of the 16-OMC standard is observed at 3.185 ppm (c). The four kinds of coffee samples spiked with 16-OMC standard had signals which slightly shifted to 3.18 ppm (a), while the position of the esterified 16-OMC is 3.17–3.18 ppm (b). This indicates that the free form of 16-OMC immediately interacts with the fatty acids in coffee.

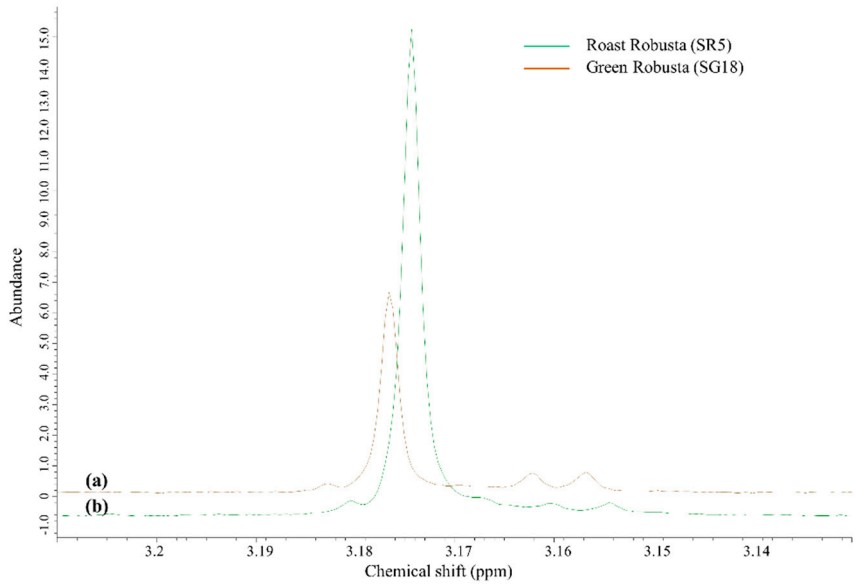

**Figure 3.** The expanded region of the 16-OMC signal for roast Robusta (**a**) and green Robusta (**b**).

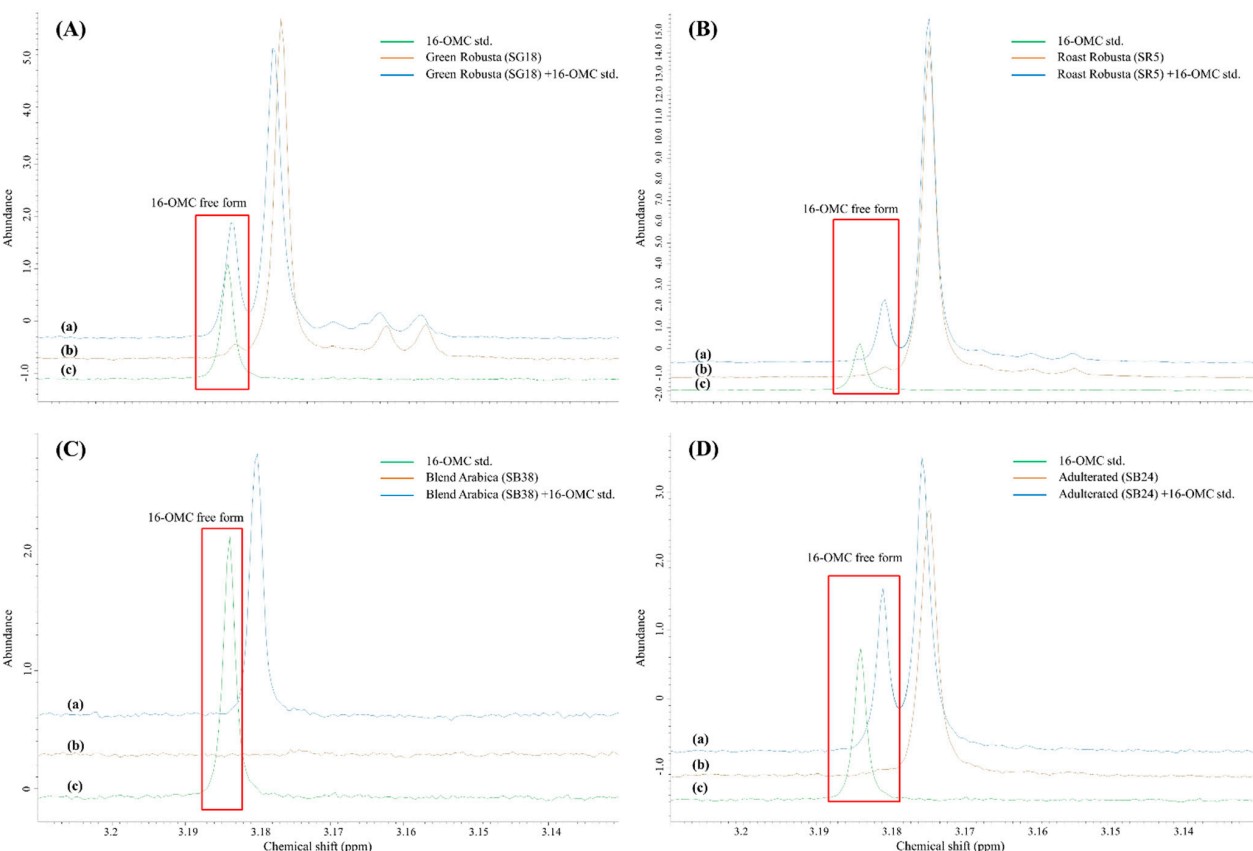

**Figure 4.** The overlap of samples with 16-OMC (**a**), coffee sample (**b**) and standard 16-OMC standard (**c**) of green Robusta (**A**), roast Robusta (**B**), blend Arabica (**C**), and adulterated sample (**D**). The signals of free form 16-OMC are indicated.

The free form and esterification of 16-OMC have been previously reported in the literature. The coffee diterpenes are mostly esterified with fatty acids, and only a small amount is present in the free form [5,7]. The diterpene esters include palmitic, linoleic, stearic, oleic, arachidic, and behenic acid in large amounts, with some minor amounts of odd-number fatty acids such as C17–C23 [21,22]. The contents of some diterpene esters in coffee brews have previously been analyzed using HPLC with a diode-array detector (DAD) and HPLC with DAD and spectral deconvolution. Regarding kahweol and cafestol esters, palmitate is the major diterpene ester [22,23]. Additionally, 16-OMC palmitate (16-OMCP) was synthesized and assigned to [1]H NMR signals [5]. However, the ratio of each diterpene ester combined with 16-OMC requires further study. Interestingly, kahweol, cafestol, and 16-OMC in both the free form and palmitate esters have been associated with human health, especially the level of serum cholesterol [21].

### 3.3. The Decomposition Products of 16-OMC

Nevertheless, two additional signals around 3.15–3.16 ppm are observed in Robusta (Figure 3). The peaks are more obvious in green Robusta than in the roasted samples. Previous work also indicates two intensive signals at 3.15–3.16 ppm present in pure Robusta [2], which were always stronger in green coffee extracts compared to roasted [1]. These extra signals are decomposition products of 16-OMC [2]. Nevertheless, the extra peaks are increased by prolonged storage time after extraction, following a decrease of 3.17 ppm signal [14]. Even though the 16-OMC standard is more stable than 16-OMC in the extracts, degradation still occurs [5]. Their study further proposed the molecular structure of the main degradation product via NMR spectroscopy and electrospray ionization mass spectrometry (ESI-MS) measurements. A possible mechanism for the degradation

of 16-OMC esters has been discussed; the additional peaks exist in the aldehyde region, suggesting that the changes involve the opening of the furan ring. Specifically, the acidic conditions in coffee and high humidity contents in green coffee may cause the formation of unstable compounds. Therefore, degradation occurs as a consequence of cell disruption during the grinding step [5,14].

According to previous studies, the free form and additional signals of 16-OMC are all integrated together within esterified 16-OMC [2,5,7]. Therefore, the signals within 3.15–3.19 ppm were all integrated in our study.

### 3.4. Validation

This quantification method was further validated for the single Arabica as a blank matrix (Table 2). Since the CV% of recovery was less than 10% in all the examined concentrations, the method was considered valid. To calculate the limit of quantification (LOQ), a signal-to-noise ratio of 10 was used. The LOQ of this method was 50 mg/kg.

**Table 2.** Recovery and precision of blank coffee spiked with 16-OMC standard.

| Concentration of Spiked 16-OMC (mg/kg) | Recovery% | CV% |
|---|---|---|
| 50 | 108.9 | 6.4 |
| 100 | 103.5 | 3.0 |
| 500 | 81.9 | 1.7 |

### 3.5. Surveillance

The surveillance results are shown in Table 1 and Supplementary Tables S1–S4. In the blend roasted coffee beans, 17 out of 47 samples claiming to include 100% Arabica had 16-OMC (Table 1) in the range of 155.74–784.60 mg/kg. All 17 samples were products from the coffee chain. Furthermore, SB10, SB18, and SB19, which were labeled as 100% coffee also produced by the coffee chain, contained 266.86–617.61 mg/kg of 16-OMC. One product from Italy labeled as Arabica (SB31) had 214.56 mg/kg of 16-OMC.

Others have studied the contents of coffee products. Schievano et al., (2014) reported 16-OMC contents in two commercial Arabica/Robusta coffee products, with concentrations of 1160 and 1537 mg/kg [7]. The range of other non-declared products is 32–736 mg/kg. Previous findings have utilized the relative integral to identify adulteration by [1]H NMR [2]. A total of six out of sixteen samples that were declared as Arabica showed traces of Robusta adulteration; the relative integral of the 16-OMC signal was between 0.017 and 0.7828. The relative integral values were smaller than their threshold of 0.075, and one sample showed the matrix effect.

There were 46 green coffee beans that were included in this study (Supplementary Table S1). For confirmation, we identified the species of green coffee beans through DNA identification. All the green coffee beans had the same species as the label, where SG9, SG18, and SG28 are Robusta, and the others are Arabica. SG42, SG45, SG46, and SG47 could not be identified because they were decaf-treated. None of the Arabica had 16-OMC content, and the 16-OMC content of Robusta were between 1005.55 and 1615.86 mg/kg.

In this study, five single roasted coffee beans were also included (Supplementary Table S2). SR1-SR4 are Arabica without the presence of 16-OMC. SR5 is Robusta and contained 3208.32 mg/kg 16-OMC. The amount of 16-OMC in SR5 and the adulterated products that are listed in Table 1 were used to estimate the adulteration ratio of roast coffee products. We speculated the commercial Arabica blends were mixed with 4.85–24.46% Robusta. Gunning et al., (2018) also estimated the prevalence of fraud is in the range 5–20% of 100% Arabica ground roast coffee products [1].

Together, the single green and roast Robusta in our study determined 16-OMC contents in the range of 1005.55–3208.32 mg/kg (Supplementary Tables S1 and S2). Others show similar 16-OMC contents of Robusta; the contents of 16-OMC in Robusta from different geographical origins are between 2236 and 1204 mg/kg [5], and three single Robusta and

three 100% Robusta products, between 1442 and 1841 mg/kg [7]. No single green or roast Arabica bean had detectable 16-OMC [5]. To summarize the results of our study, we also detected 16-OMC in all Robusta coffees and could not detect it in the Arabica coffees.

Additionally, ten ground coffee products were collected (Supplementary Table S3). All these products claimed to be 100% Arabica on the package, and 16-OMC was not found in these samples. Furthermore, four instant coffee products also lacked 16-OMC (Supplementary Table S4). However, the content of 16-OMC in instant coffee may not be a suitable marker for adulteration. Studies indicate that instant coffees contain negligible amounts of diterpenes ester content compared to other types of coffees. Industrial processing may cause the lower contents of diterpene esters in instant coffee production [22,24–26]. However, the influence of 16-OMC amount in instant coffee is still not clear. Further research is needed to confirm whether 16-OMC can be used to verify the authenticity of instant coffee.

Arabica costs double the price or more compared to Robusta. It offers a high potential for some unscrupulous traders to make economic gains by partially or wholly replacing Arabica with Robusta. It is an economically motivated fraud, usually adding a relatively high amount of cheaper material into the expensive product [16,27]. In our results, the signal of 16-OMC of Arabica does not exist but is significant in the adulterated product. Even though some Arabica coffees detected a small amount of 16-OMC [1,11,12]. However, a typical Arabica coffee contains only 1–2% of the level of combined esterified 16-OMC and 16-OMK of a typical Robusta [1]. A lot of research still used 16-OMC as the indicator to identify Arabica adulterated with Robusta [2,5,7,8,14]. For economic gain purposes, the dishonest traders will add a sufficient quantity of Robusta into Arabica coffee. It indicates the signal of 16-OMC will be elevated obviously when Arabica is adulterated with Robusta. When the LOQ of 16-OMC is above the levels in authentic Arabica, adulteration can still be measured by using 16-OMC as a marker. Therefore, we agree that 16-OMC is the suitable indicator to identify adulteration so far.

Besides 16-OMC, other indicators were also applied as markers for discrimination of Arabica and Robusta. $\Delta^5$-avenasterol can be a very adequate variable to establish the Arabica percentage in roasted coffee blends. The amount of $\Delta^5$-avenasterol in Robusta was five-fold higher than Arabica. However, it needs more complicated saponification before GC analysis [6,28,29]. Pipecolic acid betaine (homostachydrine) is the potential marker for coffee adulteration due to it being heat stable and present in higher amounts in Robusta than Arabica. [30]. Additionally, the concentration or ratio of some fatty acids could be used as indicators to assess the relative amounts of Arabica and Robusta in a coffee blend [31]. Even though the amount of difference of these indicators between Robusta and Arabica is not as significant as 16-OMC, it would however be interesting to do some further research to compare the results of 16-OMC with these indicators.

## 4. Conclusions

In this study, we quantified the 16-OMC content in coffee products using a 500 MHz NMR. Interestingly, we observed that the peak of 16-OMC free form is slightly shifted once added to the coffee sample due to the esterification. Additionally, the decomposition products of 16-OMC were observed obviously in green Robusta, but the reaction mechanism is still not fully understood. We confirmed that coffee chain adulterated Arabica with Robusta in parts of their products, which claimed to include 100% Arabica. In this case, this is an economically motivated adulteration and will not cause serious damage to health. However, it is food fraud, and can erode consumer confidence in the food industry. Some brands' products that were collected in this study were not adulterated.

**Supplementary Materials:** The following supporting information can be downloaded at: https://www.mdpi.com/article/10.3390/pr11030871/s1, Table S1: Information and 16-OMC content of single green coffee samples; Table S2: Information and 16-OMC content of single roast coffee samples; Table S3: Information and 16-OMC content of ground coffee products; Table S4: Information and 16-OMC content of instant coffee products.

**Author Contributions:** Conceptualization, D.-Y.W.; methodology, S.-Y.C. and Y.-L.W.; software, S.-Y.C. and Y.-L.W.; validation, S.-Y.C. and Y.-L.W.; formal analysis, S.-Y.C. and Y.-L.W.; investigation, S.-Y.C. and Y.-L.W.; resources, D.-Y.W.; data curation, Y.-T.L.; writing—original draft preparation, S.-Y.C.; writing—review and editing, Y.-T.L.; visualization, S.-Y.C.; supervision, Y.-T.L., Y.-M.K., S.-H.T., and D.-Y.W.; project administration, Y.-T.L., Y.-M.K., S.-H.T., and D.-Y.W. All authors have read and agreed to the published version of the manuscript.

**Funding:** This research received no external funding.

**Institutional Review Board Statement:** Not applicable.

**Informed Consent Statement:** Not applicable.

**Data Availability Statement:** The data that were presented in this study are available on request from the corresponding author.

**Acknowledgments:** We gratefully acknowledge the financial support from Food and Drug Administration, Ministry of Health and Welfare, Taiwan.

**Conflicts of Interest:** The authors declare no conflict of interest.

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
