# Peer review of "Authentication of Coffee Blends by 16-O-Methylcafestol Quantification Using NMR Spectroscopy"

_processes, doi:10.3390/pr11030871_

Round 1

Reviewer 1 Report

The present paper aimed to study the adulteration of Coffee Blends by 16-O-methylcafestol using the NMR technique. The paper was within the journal's scope and the application of NMR technology for food safety accepts is an advanced technique. The paper was well-organized and scientifically sound. However, the authors should address the following comments before publishing the paper.

·         The introduction should be strengthened by stating the drawbacks of traditional techniques and the advantages of the NMR technique over other advanced techniques along with the novelty of the present study.

·         The objective of the study should be discussed clearly at the end of the introduction section.

·         In the materials and methods section where equipment is cited, the company and equipment details should be provided.

·         Maintain the same units format throughout the manuscript

·         The discussion part is weak it should be strengthened

·         The conclusion part is missing 

Author Response

Comment 1: The present paper aimed to study the adulteration of Coffee Blends by 16-O-methylcafestol using the NMR technique. The paper was within the journal's scope and the application of NMR technology for food safety accepts is an advanced technique. The paper was well-organized and scientifically sound. However, the authors should address the following comments before publishing the paper.

Author response: Thank you! We have addressed the manuscript carefully.

Comment 2: The introduction should be strengthened by stating the drawbacks of traditional techniques and the advantages of the NMR technique over other advanced techniques along with the novelty of the present study.

Author response: Thank you for this suggestion. We mentioned about the drawbacks of traditional techniques and the advantages of the NMR technique in Line 47-54, page 2.

Comment 3: The objective of the study should be discussed clearly at the end of the introduction section.

Author response: We have added the suggested content to the manuscript in Line 57-61, page 2.

Comment 4: In the materials and methods section where equipment is cited, the company and equipment details should be provided.

Author response: Thank you for this suggestion. The detail has been corrected in Line 95-96, page 4.

Comment 5: Maintain the same units format throughout the manuscript.

Author response: Thank you for this suggestion. The unit of the 16-OMC contents from our study and from the reference is using mg/kg.

Comment 6: The discussion part is weak it should be strengthened

Author response: Thank you for this suggestion. We have added more aspects in the discussion part with tracked changes.

Comment 7: The conclusion part is missing

Author response: Thank you for pointing this out. We have corrected it.

Reviewer 2 Report

The topics covered in the manuscript are interesting and in the light of the data cited, the problem is up-to-date. However, I have a few cardinal remarks.

Arabica and Robusta coffee beans are distinguishable with the "naked eye", while samples of this type constitute the majority of the samples tested. Shouldn't the research be carried out mainly on ground and instant coffees?

Some samples showed very high SD in the 16-OMC concentrations determined. This raises basic doubts in the preparation of samples for testing. Were they representative? How many seeds were used to prepare the sample and what percentage of your products did they account for?

What is innovative about the described 1H NMR method since in the "introduction" part the authors cite six papers where it was used.

In the "introduction" section, the authors claim that 16-OMC "is present exclusively in Robusta" and then that "16-OMC or its esterified form are presenting in some Arabica coffees", this information is contradictory.

The bibliography is very poor.

No Figure 1 containing structural formulas (mentioned in the text).

Unfortunately, after reading the manuscript carefully, there is no sense that the study was properly planned and that it brings something new.

Author Response

The topics covered in the manuscript are interesting and in the light of the data cited, the problem is up-to-date. However, I have a few cardinal remarks.

Comment 1: Arabica and Robusta coffee beans are distinguishable with the "naked eye", while samples of this type constitute the majority of the samples tested. Shouldn't the research be carried out mainly on ground and instant coffees?

Author response: We think this is an excellent suggestion. This study was beginning from the scandal of 100% Arabica beans mixed with relatively cheap Robusta beans. Then, we used the objective marker 16-OMC as the indicator to detect the existence of Robusta in the products claimed 100% Arabica. In the market, the coffee bean occupies the majority proportion of the products claimed 100% Arabica. Therefore, coffee beans become the majority sample type in our study.

Comment 2: Some samples showed very high SD in the 16-OMC concentrations determined. This raises basic doubts in the preparation of samples for testing. Were they representative? How many seeds were used to prepare the sample and what percentage of your products did they account for?

Author response: Thank you for this suggestion. We believe the results are representative, most of the coefficient of variations in the sample are under 10%. We took about 30 g of coffee beans from its package, which is about 4~8% of the products according to the different sizes of each package. After we ground the beans into a powder, 0.45 g of powder was further extracted and analyzed as one replicate. We repeated the extraction at least three times.

Comment 3: What is innovative about the described 1H NMR method since in the "introduction" part the authors cite six papers where it was used.

Author response: Thank you for this suggestion. The purpose of our study is to deal with the authentication of coffee since a notable coffee chain in Taiwan was found to be mixing relatively cheap Robusta beans into products marketed as 100% Arabica. Taiwan Food and Drug Administration (TFDA) used a 500 MHz NMR to determine the content of 16-OMC in the concerned products, and find the truth of adulteration. The official report was referred as evidence to the judiciary for further action.

The innovation of our study is the esterified 16-OMC was founded and further discussed since previous studies mainly focused the discussion on the free form of 16-OMC.

Comment 4: In the "introduction" section, the authors claim that 16-OMC" is present exclusively in Robusta" and then that "16-OMC or its esterified form are presenting in some Arabica coffees", this information is contradictory.

Author response: Thank you for pointing this out. According to the references, 16-OMC is presenting exclusively in Robusta, and presenting in some Arabica coffees. However, the level of 16-OMC in Arabica is relatively much lower than in Robusta. Economic adulteration usually adds a relatively high amount of Robusta into 100% Arabica. Therefore, 16-OMC can still be used as a marker to distinguish Arabica adulterated with Robusta. The explanation is in Line 29-40, page 1.

Comment 5: The bibliography is very poor.

Author response: Thank you for this suggestion. We have added more aspects from the bibliography in the discussion part with tracked changes.

Comment 6: No Figure 1 containing structural formulas (mentioned in the text).

Author response: Thank you for pointing this out. We have added Figure 1.

Comment 7: Unfortunately, after reading the manuscript carefully, there is no sense that the study was properly planned and that it brings something new.

Author response: We appreciate the reviewer’s feedback, but we respectfully disagree. We think this study was properly planned. We started with the analytical method development and validation for the detection of 16-OMC, which is the major indicator of Arabica coffee adulteration. Then we further applied the method to identify the authentication of the concerned coffee products and confirmed the truth of adulteration. We also collected other commercial products, and none of the coffee products labeled with 100% Arabica detected the existence of 16-OMC.

The innovation part of our study is we found and further discussed the esterified 16-OMC. This aspect is rarely discussed in previous studies.

Reviewer 3 Report

Please see in the attach.

Author Response

Comments on the article "Authentication of Coffee Blends by 16-O-methylcafestol Quantification using NMR Spectroscopy" authors : Ya-Tze Lin et al,

Comment 1: Significance for FIDs. Line 93, page 4

Author response: Thank you for pointing this out. We have added the full name of FIDs, Line 97, page 4. A typical FID contains a series of n exponentially damped, complex signals, plus a background of random noise. The spectrum of NMR is obtained by Fourier transformation of the Free Induction Decay (FID) after the expansion of its time-axis data points by zero-filling, and the apparent resolution is further enhanced. FIDs can provide individual spin-spin relaxation characteristics in different phases, such as crystalline, amorphous, and interfacial.

Comment 2: References are needed for formulas (1) and (2), page 4

Author response: Thank you for this suggestion. We have added the reference for 2.5. Quantification, Line 104, page 4.

Comment 3: What does "DNA analysis" mean? line 201, page 7.

Author response: We used real-time polymerase chain reaction with the specific probe and primer to amplify the DNA of the sample. In order to identify the species of green coffee beans. Thank you for the suggestion, we addressed the contents, Line 76, page 2, and Line 209, page 8.

Comment 4: Statistical analysis is missing.

Author response: Thank you for pointing this out. We have added statistical analysis in Line 125-128, page 5.

Comment 5: Chapter 3 should be titled "Results and discussion"

Author response: Thank you for pointing this out. We have corrected it.

Comment 6: Table 1 - "Information and 16-OMC content of blend roast coffee samples" - should be included in Chapter 3, as it presents experimental results.

Author response: We mentioned Table 1 in Chapter 3, Line 194-199, page 8, and Line 217, page 8.

Comment 7: Chapter "4. Discussion" should be entitled "Conclusions".

Author response: Thank you for pointing this out. We have corrected it.

Round 2

Reviewer 2 Report

Thanks to the authors for responding to my comments. The manuscript still requires corrections, which are listed below:

1. Sample preparation information provided in response to a review should be included in the manuscript.

2. If the main innovation is the inclusion of esterified 16-OMC, this should be indicated in the abstract, purpose of the study and conclusions (or even the title).

3. The bibliography has been extended by only one item. The total number of 19 references still does not indicate a proper exploration of the subject.

Author Response

Thanks to the authors for responding to my comments. The manuscript still requires corrections, which are listed below:

Author response: Thank you very much for your valuable comments that helped us improve this manuscript.

Comment 1: Sample preparation information provided in response to a review should be included in the manuscript.

Author response: Thank you for your nice reminder. We have added the content to the manuscript in Line 92-94, page 4.

Comment 2: If the main innovation is the inclusion of esterified 16-OMC, this should be indicated in the abstract, purpose of the study and conclusions (or even the title).

Author response: Thanks for your comment. We have added the content to the manuscript in Line 19-22, page 1, Line 69-71, page 2, and Line 277-280, page 9.

Comment 3: The bibliography has been extended by only one item. The total number of 19 references still does not indicate a proper exploration of the subject.

Author response: Thanks for the comment. We have added more aspects to the bibliography in the manuscript with tracked changes.

Reviewer 3 Report

Accept in present form.

Author Response

Accept in present form.

Author response: We appreciate your previous comments that helped us improve this manuscript.